# Increase in Vascular Endothelial Growth Factor (VEGF) Expression and the Pathogenesis of iMCD-TAFRO

**DOI:** 10.3390/biomedicines12061328

**Published:** 2024-06-14

**Authors:** Gordan Srkalovic, Sally Nijim, Maya Blanka Srkalovic, David Fajgenbaum

**Affiliations:** 1Herbert-Herman Cancer Center, University of Michigan Health-Sparrow, Lansing, MI 48912, USA; 2Center for Cytokine Storm Treatment & Laboratory, Perelman School of Medicine, University of Pennsylvania, Philadelphia, PA 19104, USA; sally.nijim@pennmedicine.upenn.edu (S.N.); davidfa@pennmedicine.upenn.edu (D.F.); 3School of Medicine, Lake Erie College of Osteopathic Medicine, Erie, PA 16509, USA; msrkalovic98731@med.lecom.edu

**Keywords:** VEGF, TAFRO, i-MCD, IL-6, mTOR, signaling pathways

## Abstract

TAFRO (thrombocytopenia (T), anasarca (A), fever (F), reticulin fibrosis (F/R), renal failure (R), and organomegaly (O)) is a heterogeneous clinical subtype of idiopathic multicentric Castleman disease (iMCD) associated with a significantly poorer prognosis than other subtypes of iMCD. TAFRO symptomatology can also be seen in pathological contexts outside of iMCD, but it is unclear if those cases should be considered representative of a different disease entity or simply a severe presentation of other infectious, malignant, and rheumatological diseases. While interleukin-6 (IL-6) is an established driver of iMCD-TAFRO pathogenesis in a subset of patients, the etiology is unknown. Recent case reports and literature reviews on TAFRO patients suggest that vascular endothelial growth factor (VEGF), and the interplay of VEGF and IL-6 in concert, rather than IL-6 as a single cytokine, may be drivers for iMCD-TAFRO pathophysiology, especially renal injury. In this review, we discuss the possible role of VEGF in the pathophysiology and clinical manifestations of iMCD-TAFRO. In particular, VEGF may be involved in iMCD-TAFRO pathology through its ability to activate RAS/RAF/MEK/ERK and PI3K/AKT/mTOR signaling pathways. Further elucidating a role for the VEGF-IL-6 axis and additional disease drivers may shed light on therapeutic options for the treatment of TAFRO patients who do not respond to, or otherwise relapse following, treatment with IL-6 targeting drugs. This review investigates the potential role of VEGF in the pathophysiology of iMCD-TAFRO and the potential for targeting related signaling pathways in the future.

## 1. Introduction

Castleman disease (CD) includes a group of rare, and still inadequately understood, lymphoproliferative diseases that can be fatal, if inadequately treated. While about 4300–5200 individuals are diagnosed annually with a form of CD in the United States [1], significant variability exists in prognosis, clinical outcome, and management strategies between the various CD clinical subtypes. As such, research efforts have been made to characterize mechanistic commonalities and distinctions. 

Initially described in the 1950s by Benjamin Castleman as a condition involving unicentric lymphadenopathy with histopathological findings of angiofollicular lymph node hyperplasia present in a localized area [2,3], CD has been since described to involve multicentric lymphadenopathy, leading to the first major classification of the disease into unicentric CD (UCD) and multicentric CD (MCD) [4]. UCD manifests as localized lymphadenopathy and is typically successfully treated with surgery [5]. MCD is often characterized by progressive flares of systemic inflammation, polyclonal lymphoproliferation, and hypercytokinemia associated with multi-system organ failure, which can lead to death [4]. The etiologic diversity of MCD, and its impact on treatment, has influenced the subsequent classification of MCD into human herpesvirus 8 (HHV8)-associated MCD (HHV8-MCD) and HHV8-negative/idiopathic MCD (iMCD) [6]. An estimated one-half of MCD cases are due to uncontrolled HHV8 infection, especially among immunocompromised individuals such as patients chronically infected with human immunodeficiency virus (HIV) [7,8]. The remaining MCD cases are of unknown, idiopathic etiology (iMCD). Several hypotheses have been posited regarding disease etiology, including viral-, autoinflammatory-, autoimmune-, or neoplastic-related mechanisms [7,8]. However, a non-HHV8 infectious cause of iMCD has since become less favored in the context of two studies that were unable to identify infectious agents connected with CD using orthogonal deep sequencing platforms to compare sequences to known viruses, bacteria, fungi, and parasites [9,10]. MCD can also co-occur in the context of a para-neoplastic syndrome, POEMS (polyneuropathy, organomegaly, endocrinopathy, monoclonal plasma cell disorder, skin changes), which requires a distinct treatment regimen and is comparatively more successfully managed.

iMCD can present at any age and is slightly more frequent among men than women [11,12]. While the etiology of iMCD is not currently known, the symptoms are a result of a cytokine storm, often driven by interleukin 6 (IL-6) and other pro-inflammatory cytokines [6]. Presentation is variable, spanning from mild lymphadenopathy and systemic inflammation to more severe cases involving intense inflammation, hepatosplenomegaly, capillary leak syndrome with anasarca, pleural effusion and ascites, organ failure, and death [6,11]. The hypercytokinemia, often driven by IL-6, is thought to be responsible for the clinical manifestations and end-organ damage. In addition to characteristic clinical features, iMCD patients also demonstrate a variety of histopathological features that can also be seen with HHV8-MCD and UCD [13,14,15]. 

The differences in clinical presentation, acuity, and morbidity between iMCD patients has led to the further classification of patients into iMCD-TAFRO and iMCD-not otherwise specified (iMCD-NOS). iMCD-TAFRO patients typically represent approximately 10–40% of iMCD, depending on the cohort [16], and are clinically characterized by thrombocytopenia (T), anasarca (A), fever (F), reticulin fibrosis (F/R), renal failure (R), organomegaly (O), and normal or only slightly elevated immunoglobulin levels. iMCD-TAFRO patients tend to primarily demonstrate hyper-vascular or mixed histopathology in lymph nodes, as well as myelofibrosis and megakaryocyte hyperplasia in the bone marrow [1]. iMCD patients with this constellation of features were first described as having “TAFRO syndrome” (iMCD-TAFRO) in 2010 by Takai et al. [17], but iMCD-TAFRO cases have been reported for decades. While most were initially described among Japanese case studies, and the first case in Europe was initially published only 10 years ago [18], it is clear that TAFRO cases are not geographically or ethnically restricted [19]. iMCD-NOS has recently been further subtyped into idiopathic plasmocytic lymphadenopathy (IPL), which is characterized by polyclonal hypergammaglobulinemia and sheets of plasma cells in lymph nodes [20]. Of the three iMCD subtypes, iMCD-TAFRO has the poorest prognosis. Compared to patients with iMCD-NOS (Figure 1) without IPL, who had a 3 year survival of 87.2% in a retrospective national study of 1634 CD patients in China, patients with iMCD-TAFRO had a 3 year survival of 65.7%. A particularly sharp increase in mortality of iMCD-TAFRO patients is seen within the first 6 months. Due to the increased morbidity associated with iMCD-TAFRO cases, understanding and distinguishing its mechanistic features is of critical clinical importance.

## 2. TAFRO Syndrome

TAFRO syndrome is a heterogenous clinical syndrome—reported in the context of variable infectious, malignant, and rheumatological diseases, in addition to iMCD [21]. However, iMCD has been determined to be one of the primary diseases in which TAFRO syndrome is observed, with validated diagnostic criteria proposed for both TAFRO syndrome (without iMCD) and iMCD-TAFRO [19]. Apart from an aggressive clinical course, iMCD-TAFRO has been observed to have distinct phenotypic markers compared to iMCD-NOS, including very low platelet count (compared to elevated platelet counts in iMCD-NOS), elevated alkaline phosphatase without hyperbilirubinemia, elevated transaminases, and a notable lack of polyclonal hypergammaglobulinemia [21]. However, these phenotypic features of iMCD-TAFRO are non-specific and can be found in a number of other conditions, making accurate diagnosis of TAFRO difficult. In addition to the broader diagnostic challenges of iMCD, including lack of specific diagnostic biomarkers and poor pathophysiologic understanding, iMCD-TAFRO diagnosis is hindered by rapid clinical deterioration, significant thrombocytopenia, and particularly small volume lymphadenopathy which makes lymph nodes biopsies challenging [21,22].

Beyond TAFRO’s unifying clinical features, its overlap with iMCD and heterogenous disease associations has made understanding iMCD-TAFRO’s mechanistic underpinnings elusive. As such, efforts have been made to further define this condition. In an effort to clarify mechanism and disease classification, open research questions have included whether TAFRO syndrome represents the most “extreme clinical presentation” of iMCD along a spectrum, or if TAFRO syndrome should be regarded as a distinct entity overlapping with iMCD with unique laboratory, clinical, and pathophysiological implications compared to other CD subtypes. 

Due to disease heterogeneity, absence of specific markers, ambiguous pathophysiology, rapid clinical onset and deterioration, and lack of consensus regarding TAFRO syndrome classification, efforts were made to reconcile disparate definitions and diagnostic criteria. In 2021, a group of experts published a validated international definition of the TAFRO clinical subtype of iMCD [21] based on a systematic literature review which included 65 case reports, including 75 patients with iMCD-TAFRO. 54 published cases with iMCD-TAFRO, TAFRO syndrome with possible iMCD without lymph node biopsy, or non-iMCD TAFRO syndrome were used to develop a definition of iMCD-TAFRO validated on the ACCELERATE Natural History Registry (NCT02817997). Criteria were compared to 69 reviewed, expert-confirmed pathology samples and cases of iMCD to guarantee external validity [19]. The result of this effort was an international definition of iMCD-TAFRO (Table 1). The current definition and diagnostic criteria for TAFRO syndrome and iMCD-TAFRO are presented in Table 2.

Within the context of consensus diagnostic definitions for TAFRO and iMCD-TAFRO, recent case reports and literature reviews of patients have collectively seemed to suggest that vascular endothelial growth factor (VEGF)—or, alternatively, the VEGF-IL-6 axis which has been suggested to reinforce expression through reciprocal signaling between bone marrow and lymph nodes, rather than IL-6 as a single cytokine—may be a driver of iMCD-TAFRO pathophysiology [23,24]. In this review, we seek to discuss and evaluate existing evidence surrounding the possible role of VEGF in iMCD-TAFRO in the context of VEGF’s complex biological interplay with multiple signaling pathways.

## 3. Vascular Endothelial Factors, Biology, Mechanism and Regulation of Receptor Signaling

Vascular endothelial growth factors (VEGFs) include a class of signaling proteins—vertebrate VEGFs A–D, placenta growth factor (PlGF), parapoxvirus VEGF-E, as well as snake venom VEGF-F [25]—that are broadly expressed, structurally related dimeric molecules with critical roles in vasculogenesis and angiogenesis. By exerting mitogenic and anti-apoptotic effects on endothelial cells, VEGFs play vital roles in regulating vascular permeability and promoting cell migration that is essential to the function and development of multiple organ systems, including the central nervous system (CNS), kidney, lung, and liver [25,26].

VEGFs bind with high affinity to the receptor tyrosine kinases (RTKs) VEGFR1, VEGFR2, and VEGFR3. Of the VEGFR isoforoms, VEGFR1 and VEGFR2 are expressed primarily on vascular endothelial cells, whereas VEGFR3 is mainly located on lymphatic endothelial cells. As such, while VEGF-C/D and their receptor VEGFR3 are primarily involved in lymphangiogenesis, VEGF-A activates VEGFR1 and VEGFR2 and primarily exerts regulatory effects on the cell migration of macrophage and endothelial cell lineages, vasculogenesis, and vascular permeability—phenomena primarily observed in iMCD-TAFRO. Although VEGF-A binds to VEGFR1 with high affinity (10 pM), the induction of VEGFR1 phosphorylation is weak, and VEGFR1 is not primarily thought to be a receptor transmitting a mitogenic signal, but, rather, a ‘decoy’ receptor [27]; however, a motif in the juxtamembrane region of VEGFR1, but not VEGFR2, represses PI3K activation and EC migration [28]. As such, VEGFR2 is thought to act as the primary signaling VEGFR in blood vascular endothelial cells. However, recent studies have also provided evidence for a non-mitogenic function of VEGFR1 in the liver: VEGFR1 activation resulted in the paracrine release of hepatocyte growth factor and other hepatotropic molecules including IL-6 [29].

Given its dynamic importance in multiple organ systems, VEGF-A expression is tightly regulated at transcriptional and translational levels via a network of hypoxic and non-hypoxic-dependent mechanisms. Hypoxic responses in cells and tissues are largely mediated by the family of hypoxia-inducible factor (HIF) transcription factors, which play an integral role in cellular adaptation to low oxygen availability [30] and ultimately activate many target genes, including VEGFA [31]. While one of the best-characterized regulators of VEGF release is hypoxia, several non-hypoxic and non-HIF-mediated regulatory mechanisms exist [32]. In fact, inhibition of PI3K/AKT/mTOR signaling, which has been implicated in iMCD-TAFRO patients, strongly downregulates the hypoxic induction of VEGF [33]. Evidence of PI3K/AKT/mTOR-mediated regulatory mechanisms have been demonstrated in multiple pre-clinical and clinical contexts for various VEGF isoforms. For example, mTOR inhibition with sirolimus has been demonstrated to lower VEGF-A levels in iMCD [34] and VEGF-D expression longitudinally in lymphangioleiomyomatosis [35]. Further, signaling cascades such as the RAS pathway, which cross-talk with PI3K/AKT/mTOR signaling, exert regulatory effects on VEGF. In a non-pathological setting, activated KRAS can directly augment VEGF gene expression via HIF-1 alpha or modulate other transcription factors to induce similar effects indirectly. In a pathological context, one of the HIF-independent, hypoxia-dependent mechanisms for the regulation of VEGF expression is the KRAS oncogene-mediated activation of VEGFA, whereby mutated KRAS augments VEGF-A induction. This was additionally observed in wild-type and HIF1A-knockdown colon cancer cells [36].

Downstream of aforementioned regulatory pathways and, similar to other tyrosine kinase receptors within this class, VEGF-A binds to its cognate VEGF receptor to induce receptor homodimerization or heterodimerization, resulting in activation and autophosphorylation of tyrosine residues in the receptor intracellular domains [37]. Phosphorylated amino acid residues constitute binding sites for the mediation of diverse pathways, including STAT1 and STAT3 signaling, the PI3K-AKT pathway and small GTPases, as well as the SRC family of cytoplasmic tyrosine kinases involved in cell shape, integration, and polarization. Other downstream pathways also include the PLcy-ERK1/2 pathway, which is crucial for vascular development. Aforementioned pathways mediate rapid responses, including immediate changes in vascular permeability, and longer-term responses requiring gene regulation, such as endothelial cell survival, migration, and proliferation.

During downstream signaling, with specific attention to the PI3K/AKT/mTOR pathway, AKT serine/threonine kinases AKT1–3 act on a wide range of substrates and influence many biological processes, including cell survival, proliferation, and apoptosis [38,39]. Activation of AKT requires binding of its plextrin homology (PH) domain to the lipid second messenger phosphatidylinositol-3,4,5-trisphosphate (PIP3) generated by PI3K. AKT1 is the predominant isoform involved in the regulation of pathological and adult angiogenesis, as well as vascular maturation and metabolism by the mTOR complex 2 (Figure 2). By contrast, AKT2 knockouts lack an overt phenotype [40,41,42,43]. VEGFR2 lacks a binding site for the SH2 domain containing the p85 subunit of PI3K and activates PI3K indirectly, either by SRC and VE-cadherin or by AXL [44,45].

Other non-canonical forms of VEGFR activation outside of dimerization and phosphorylation pathways include non-VEGF-ligand-dependent activation via shear-stress, resulting in PECAM1 phosphorylation and PI3K activation. Given the absence of in vivo models, assessments relating to shear stress have not been explored in the context of iMCD-TAFRO, and the tortuous vessels observed on pathology are currently hypothesized to be a byproduct of VEGF production. However, it is possible that this mechanism may help perpetuate hypervascularity and survival of vascular endothelial cells in iMCD-TAFRO, analogous to other biological contexts [47]. In the context of tight VEGF-A regulation, minimal disturbances in homeostasis are pathological. Indeed, both 50% reductions and excess of VEGF in experimental models prove detrimental to organisms [48]. Complex networks, including aforementioned signaling pathways, enable a critical balance of VEGF-mediated processes.

## 4. The Role of VEGF in Pathophysiology of TAFRO Syndrome

Elevated IL-6, as well as increased PI3K/AKT/mTOR signaling, have both been demonstrated to be disease-driving hallmarks in a subset of iMCD-TAFRO cases [1]. However, 50–66% of patients do not respond to the only FDA-approved therapy for iMCD, siltuximab, which targets IL-6 [49]. This highlights the necessity of exploring additional targetable pathways involved in pathogenesis of IL-6 blockade-refractory iMCD and enhancing current pathophysiologic understanding of iMCD-TAFRO. Lee et al. (2020) recently conducted a systematic literature review of 66 iMCD patients regarding the role of VEGF in the pathogenesis of TAFRO syndrome and found support for a role of VEGF-A in TAFRO, with fibroblastic reticular cells (FRCs) as the likely pathological VEGF-A-expressing cells [23]. The results yielded from this array of patient cases supported a correlation between VEGF-A and TAFRO disease activity, but the current source of VEGF-A production in iMCD is not known and may be associated with plasma cell proliferation within lymph nodes and/or from fibroblastic reticular cells in bone marrow [23,50].

In studies conducted by Fajgenbaum et al., hallmarks of iMCD-TAFRO flare included elevated CD8^+^ T-cells and elevated VEGF levels [34]. Using lymph node tissue and blood samples from 3 IL-6 blockade-refractory iMCD-TAFRO patients, increased VEGF-A levels, CD8^+^ T-cell activation, and PI3K/AKT/mTOR activity were detected during disease flare [34]. Analyses of Patient 1’s serum, drawn in the months preceding a fifth flare [34], identified VEGF-A as the sole marker, other than sIL-2R alpha, that significantly rose above the upper limit of normal, peaking at 3-fold during flare (Figure 3). A similar pattern was observed during the two previous flares in the same patient. These initial analyses of cyclical VEGF elevations during flare, and prior to flare, were first supportive of a pathophysiologic role for VEGF mediated by mTOR specifically in iMCD-TAFRO. None of the other 11 clinically measured serum inflammatory markers demonstrated a consistent upward trend prior to flare onset. Phenotypic clinical manifestations in these patients associated with elevated VEGF-A—including cherry hemangiomatosis, capillary leak syndrome, and lymph node hypervascularization—are also consistent with this.

Serum-based proteomics studies conducted using the same patient’s samples prior to flare on a 315-analyte platform also detected that sIL-2Rα and VEGF-A were significantly greater at the onset of flare compared to remission. Such longitudinal changes in sIL-2Rα and VEGF-A in non-flare patients compared to healthy controls are also supportive of a role for T-cell activation and VEGF-A, with increased PI3K/AKT/mTOR activity linking the two in iMCD for the first time (Figure 4). Compared to Patient 1 from this cohort, the other two patients similarly exhibited elevated VEGF-A, with differences that were greater than 2 and 20 times the upper limit normal (ULN). All three patients demonstrated clinicopathologic features associated with elevated VEGF-A, such as hypervascularized lymph nodes and capillary leak syndrome.

Since these initial studies, other serum-based proteomics studies on larger iMCD cohorts have been conducted, including Pierson et al. (2021) [51] and Pierson et al. (2022) [52]. Following extrapolation of TAFRO status from patient platelet counts, these studies reported increases in VEGF during flare in iMCD-TAFRO patients compared to healthy and disease controls [51,52]. In fact, VEGF was found to be one of the top four serum markers elevated within both iMCD and iMCD-TAFRO flares. Other studies have reported similar findings. Iwaki et al. found VEGF-A serum concentrations to be significantly higher in iMCD-TAFRO and iMCD-NOS compared to healthy controls [24]. Due to the lack of IL-6-associated thrombocytosis and polyclonal hypergammaglobulinemia in iMCD-TAFRO phenotypes, their work has additionally suggested that elevated serum IL-6 may not be a sole pathological driver of hypercytokinemia in patients with iMCD-TAFRO. Other case studies, including Oka et al. (2018), have recorded elevated VEGF-A in two patients with iMCD-TAFRO during flare, which similarly returned to normal levels with successful treatment [53]. Zhou also reported elevated VEGF-A in all five patients diagnosed with iMCD-TAFRO, with levels extremely elevated (>800 pg/mL) in four out of five patients [54]. In a systematic literature review including 128 patients, Liu et al. found increased concentrations of VEGF in 16/20 patients reporting that cytokine. While it is not clear if any of those patients with reported VEGF levels were iMCD-TAFRO patients, 19% of all reported patients in this paper had TAFRO characteristics [11].

VEGF has also been implicated in renal and peri-renal injury in iMCD-TAFRO patients. Observed cases of iMCD-TAFRO renal histopathology have been consistent with membranoproliferative glomerulonephritis (MPGN; 42%) and thrombotic microangiopathy-like (TMA-like; 58%) patterns, but with the latter absent of its characteristic biological signs or fibrin thrombi in glomerular capillaries and arterioles. In a review of 19 patient cases of iMCD-TAFRO [55], serum IL-6 and VEGF were elevated in all but one case, where lab values were accessible [56]. While the pathogenesis of renal injury in iMCD is unclear, it has been hypothesized that IL-6 and VEGF may be involved in renal complications in the same manner they may be involved in the pathogenesis of iMCD.

Podocytes in the glomerular endothelium have been found to produce VEGF, which stimulates endothelial cell fenestrations that maintain the permeability of the glomerular filtration barrier [57,58]. Reduced glomerular VEGF expression results in loss of these fenestrations, leading to microvascular injury and TMA, even if elevated circulating VEGF is observed [58]. A possible mechanism underlying this process could be a negative feedback mechanism, wherein glomerular-specific expression is reduced as a result of increased circulating VEGF during the cytokine storm characteristic of iMCD-TAFRO. Accordingly, elevated serum VEGF (and IL-6) would be expected to suppress glomerular VEGF expression, producing microvascular injury [59]. This is supported by case studies of iMCD-TAFRO in which VEGF glomerular staining was observed to be lower in at least a portion of patients with small-vessel lesions—in direct correlation with CD disease activity. As such, particularly for TMA-type glomerular injury, a negative feedback mechanism may account for the contrasting combination of increased circulating VEGF and reduced local levels [60]. Although this hypothesis explains some of the findings in iMCD-TAFRO patients with renal injury, additional investigation is required, particularly since lowered local VEGF expression is not observed in all patients. Other cytokines, such as IL-6 or PDGF-B, may also be involved in renal injury in these patients [54]. Macrophages, key producers of both IL-6 and VEGF, have been identified as cellular infiltrates in the kidneys of iMCD-TAFRO patients [55,61]. In addition, in small cohort studies comparing TAFRO and iMCD non-TAFRO patients (iMCD-NOS), treatment—often with an IL-6 inhibitor—resulted in complete recovery of renal findings in all patients [62]. However, this finding was not observed in iMCD-NOS. Given the infiltration of a consistent cell type on pathology and the pathway’s effector role in podocyte maintenance, IL-6-VEGF-axis-induced glomerular microangiopathy may play a role in developing acute kidney injury in TAFRO syndrome.

The particular role that VEGF, as well as its feedback mechanisms, play in the context of renal TMA are unclear [63]. VEGFR2 downstream effects via autocrine or paracrine signaling may induce glomerular damage as a result of VEGF overexpression [64]. Nagayama et al., for example, did not find a difference in glomerular VEGF-A staining in podocytes between control and TAFRO cases, but did find significant increases in VEGF-A in the renal cortex in TAFRO cases [63]. A relationship between VEGF renal disturbances and glomerular microangiopathic injury is further supported by data on systemic anti-VEGF therapy in six patients, which induced TMA-like glomerulopathy. Murine models, including podocyte-specific deletion of VEGF, also developed TMA-like glomerular injury patterns [57]. In fact, both surplus and insufficient VEGF in the glomerulus have been suggested to lead to glomerular damage [63]. While the aforementioned evidence has been supportive in the context of TMA-like renal injury, it has been hypothesized that alternative lesions, including MPGN, may be related to chronicity. In collections of case studies, patients observed to have TMA-like renal injury patterns tended to have earlier renal biopsies relative to those patients who had an MPGN pattern of injury, suggesting that MPGN lesions may result from chronic TMA lesions [64]. Peri-renal injury patterns specific to iMCD-TAFRO patients have also been observed, including adrenal hemorrhage and adrenal ischemia [65]. Further research is required regarding a potential concurrent role for VEGF in peri-renal injury manifestations.

## 5. Conclusions and Future Research Directions

Despite evidence of a role for VEGF in the pathophysiology of iMCD-TAFRO, most existing data are based on limited patient cohorts or systematic literature reviews, as well as preclinical data. Additional future research focused on this important signaling protein’s role is necessary. No cell lines or animal models are currently available for high-throughput target identification in IL-6 blockade refractory iMCD-TAFRO patients [34]. However, recent research is suggestive of potential targetable pathways. Clinically important correlations between VEGF, VEGF-R and PI3K/AKT/mTOR signaling pathways have already been established and some have been targeted in iMCD-TAFRO. The administration of sirolimus, a potent mTOR inhibitor, significantly attenuated CD8⁺ T cell activation and decreased VEGF-A levels, inducing clinical benefit in all three patients with durable and ongoing remission [49]. A clinical trial with sirolimus in previously treated iMCD patients is ongoing (NCT03933904).

However, sirolimus does not appear to be uniformly effective in IL-6-therapy-refractory cases, and the most severe cases such as iMCD-TAFRO often require multi-agent treatment. While the role VEGF may play in pathogenesis of TAFRO is ambiguous, it is evident that the VEGF-IL-6 axis interfaces with several newly discovered targetable signaling pathways, including JAK-STAT signaling [35]. The likely efficacy of targeting other signaling pathways interfacing with VEGF expression is unclear but requires further investigation. Other VEGFR downstream pathways such as RAS/RAF/MEK/ERK could be investigated, particularly if cell lines, animal models, or datasets for iMCD-TAFRO are developed. Targeting VEGFR tyrosine kinase (TK) with multiple or specific kinase inhibitors presents another opportunity considering the number of already-approved drugs (sorafenib, suntinib, vandetanib, axitinib, cabozantinib, regorafenib, apatinib, nindetinib, lenvatinibanlotinib, fruquintinib, and tivozantinib) (Figure 5). These drugs have different targets, and some are specific for VEGFR 1/2/3, such as axitinib, fruquintinb, and tivazantinib, while others target other tyrosine kinases besides VEGFR, such as c-Kit, FLT3, PDGFR-α/β, RET, FGFR-1/3, and others [46].

Complicated interactions between subclasses of VEGF receptors and the activation of multiple downstream pathways make targeting specific receptors a challenging goal, and multiple patients will likely require treatment from a diversity of therapies. This also highlights the necessity of precise predictive biomarkers and an enhanced target understanding before translation to clinical practice. Other possible approaches in the treatment of iMCD-TAFRO require attention. Although based on case reports, some of them do present proof of principle [53]. Additional biomarkers unrelated or distantly related to VEGF could also be considered as possible targets [24,66]. Additional research is necessary to increase diagnostic and therapeutic understanding of the iMCD-TAFRO mechanisms.

While investigating VEGF can shed additional light on diagnostic and therapeutic targets if the nuances of its role are established, additional questions remain, including its influence on the development of renal injury during iMCD-TAFRO. Whether VEGF plays a protective or damaging role in the kidneys of patients with iMCD-TAFRO is controversial. Given its complex regulatory mechanisms and preclinical evidence, VEGF homeostasis is likely crucial for preserving renal function and factors disturbing its homeostasis are likely to produce renal injury. Additional study regarding biomarkers of VEGF activity, particularly in renal injury associated with iMCD-TAFRO, could shed light on targets for VEGF-active drugs if additional evidence is generated in favor of a pathologic role for VEGF.

## Figures and Tables

**Figure 1 biomedicines-12-01328-f001:**
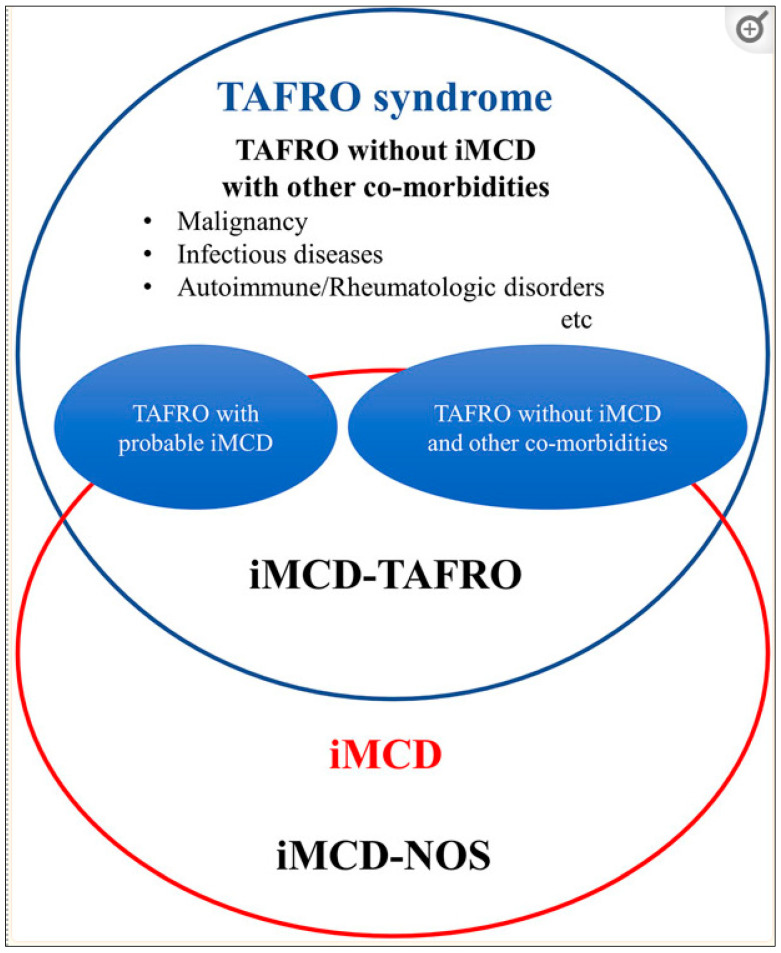
Concepts of TAFRO syndrome and iMCD-TAFRO. TAFRO syndrome is a heterogenous clinical entity with a constellation of non-specific clinical symptoms including thrombocytopenia (T), anasarca (A), fever (F), reticulin fibrosis or renal insufficiency (R), and organomegaly (O). Due to its heterogeneity, TAFRO syndrome includes various clinical conditions such as malignancies, rheumatologic disorders, infections, and POEMS syndrome. The figure conceptualizes five different classifications related to TAFRO syndrome and iMCD-TAFRO. The present study included cases with iMCD-TAFRO, TAFRO with possible iMCD without lymph node biopsy and other co-morbidities (TAFRO with possible iMCD), and TAFRO without histologically proven iMCD and other co-morbidities (TAFRO without iMCD and other co-morbidities). Attention needs to be paid not to confuse TAFRO syndrome and iMCD-TAFRO. Abbreviations: iMCD-TAFRO, TAFRO clinical subtype of idiopathic multicentric Castleman disease; iMCD-NOS, idiopathic multicentric Castleman disease not otherwise specified. Reproduced with permission from Nishimura, Y., Fajgenbaum D.VC., Pierson S.K. et al. Validated International Definition of the TAFRO clinical subtype of idiopathic multicentric Castleman disease. Am J Hematol, 2021, 96(10), 1241–1252 [21].

**Figure 2 biomedicines-12-01328-f002:**
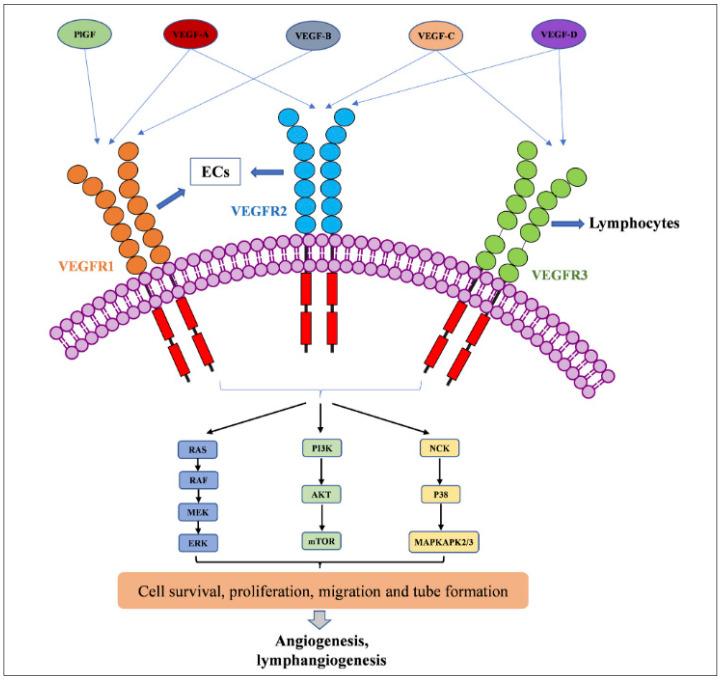
VEGFR activation by the VEGF family of growth factors and resulting downstream signaling pathways to angiogenesis and lymphangiogenesis. Family of glycoproteins including placental growth factors (PIGF) and VEGF-A, -B, -C, and -D bind to extracellular immunoglobulin (Ig) loops of VEGFR-1, -2, and-3. VEGFR-1 and -2 are mostly expressed on endothelial cells (ECs) and some cancer cells. This binding activates downstream signaling pathways, including RAS-RAF-MEK-ERK, as well as PI3K-AKT-mTOR and NCK-MAPK 2/3. Reproduced with permission from Wang, L., Liu, W-Q., Broussy, S. Han, B. and Fang H. Recent advances of anti-angiogenic inhibitors targeting VEGF/VEGFR axis. Front Pharmacol, 2024, 14, 1307860, doi: 10.3389/fphar.2023.1307860 [46].

**Figure 3 biomedicines-12-01328-f003:**
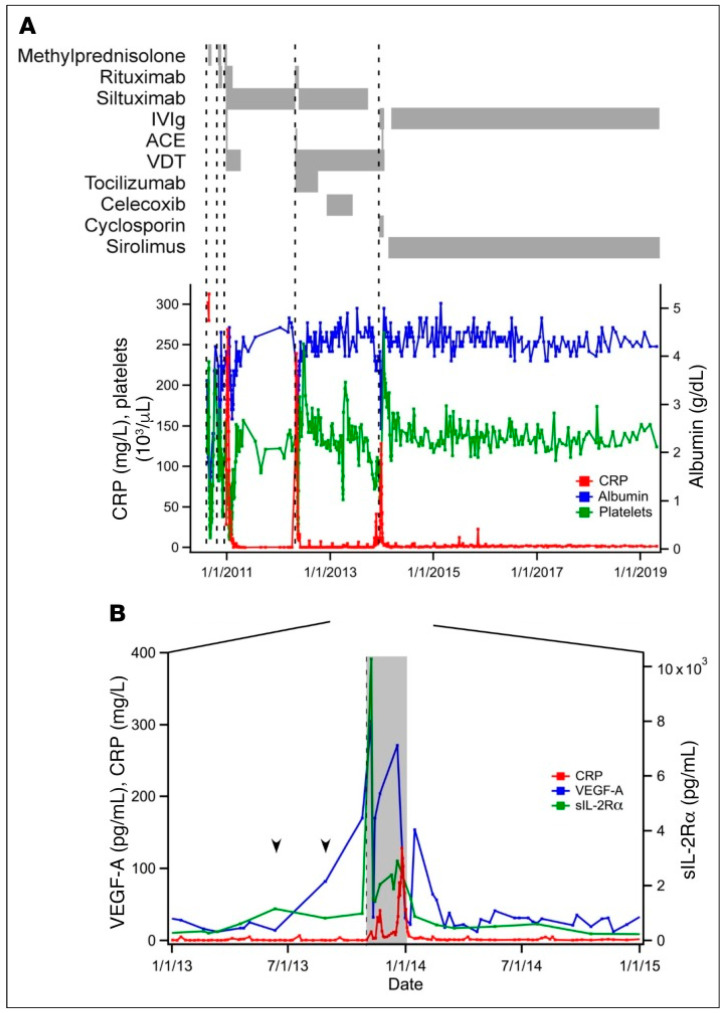
Clinical course and elevation of VEGF-A and sIL-2Rα prior to disease flare for iMCD-1. (**A**) Select laboratory values, dates of initiation of disease flares (dotted vertical lines; defined by hypoalbuminemia (<3.5 g/dL), elevated CRP (>10 mg/L), anemia (hemoglobin < 13.5 g/dL), renal dysfunction (creatinine > 1.3 mg/dL), constitutional symptoms, and fluid accumulation), and treatment regimens administered throughout iMCD-1’s disease course (*n* = 1). CRP closely parallels disease status. IVIg, intravenous immunoglobulin; ACE, doxorubicin (adriamycin)-cyclophosphamide-etoposide; VDT, bortezomib (velcade)-dexamethasone-thalidomide; CRP, C-reactive protein. (**B**) Serum levels of sIL-2Rα (normal < 1022 pg/mL) and VEGF-A (normal < 86 pg/mL) from 1 year before to 1 year after iMCD-1’s fifth disease flare (onset indicated by dotted vertical line; duration by shaded region), with CRP included for reference. Arrows indicate when sIL-2Rα and VEGF-A rose above the ULN. Reproduced with permission from Fajgenbaum D.C., Langan R-A., Japp A.S. et al. Identifying and targeting pathogenic PI3K/AKT/mTOR signaling in IL-6 blockade-refractory idiopathic multicentric Castleman disease. J Clin Invest, 2019, 129(10), 4451–4463 [34].

**Figure 4 biomedicines-12-01328-f004:**
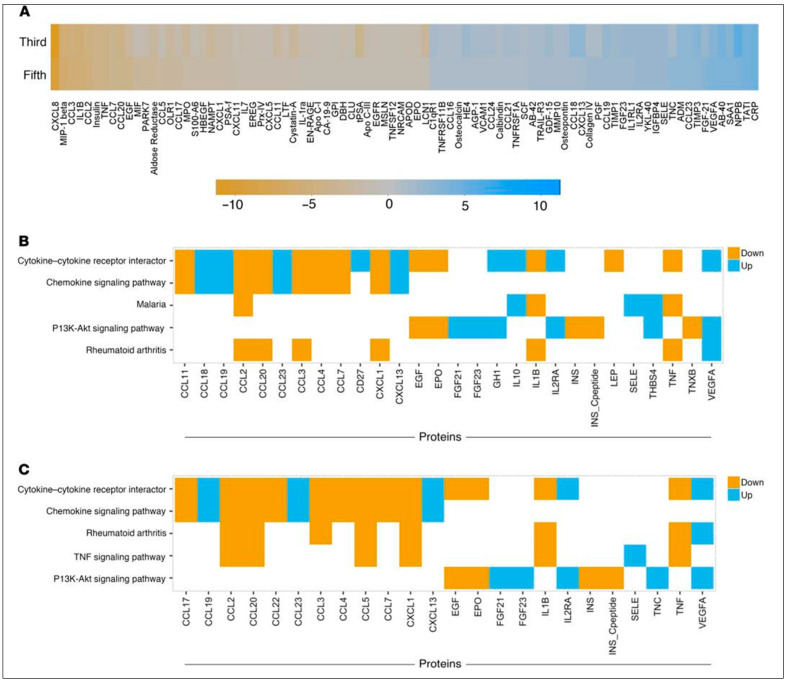
Serum proteomics and pathway analyses identify VEGF-A, sIL-2Rα, and PI3K/Akt/mTOR signaling as candidate therapeutic targets for iMCD-1. (**A**) Heatmap of the analytes whose levels increase (blue) or decrease (orange) by at least 2-fold in the same direction between flare and remission for iMCD-1’s third and fifth flares, as measured by Myriad RBM DiscoveryMAP (*n* = 1). Analytes are presented in ascending order from left to right based on the log_2_ (flare/remission) fold-change at the fifth flare, compared with remission. Key provides the color intensity for a given fold-change. (**B**,**C**) Enrichment analysis, using Enrichr, of Myriad RBM DiscoveryMAP gene sets for metabolic pathways for iMCD-1. Results of the top five enriched gene sets (FDR < 0.01, rank ordered by combined score) from the (**B**) third flare and (**C**) fifth flare when proteins with log_2_ (flare/remission) greater than two were analyzed for KEGG pathway gene sets. Colored cells represent gene members in specific pathways that were found to be greater than 4-fold up (blue) or 4-fold down (orange) during flare compared with remission. Reproduced with permission from Fajgenbaum D.C., Langan R-A., Japp A.S. et al. Identifying and targeting pathogenic PI3K/AKT/mTOR signaling in IL-6 blockade-refractory idiopathic multicentric Castleman disease. J Clin Invest, 2019, 129(10), 4451–4463 [34].

**Figure 5 biomedicines-12-01328-f005:**
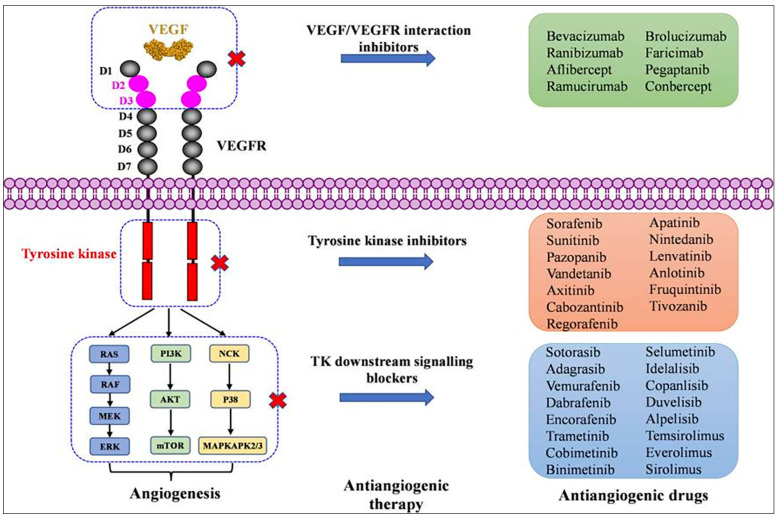
Marketed drugs targeting the VEGF/VEGFR axis and its downstream signaling pathway. Figure presents drugs marketed around the world with principal mode of action grouping them into ones that act on interaction of VEGF and VEGF-R, those that inhibit tyrosine kinase and those that interact or block downstream signaling through one of the three major pathways. Reproduced with permission from Wang, L., Liu, W-Q., Broussy, S. Han, B. and Fang H. Recent advances of anti-angiogenic inhibitors targeting VEGF/VEGFR axis. Front Pharmacol, 2024, 14, 1307860, doi: 10.3389/fphar.2023.1307860 [46].

**Table 1 biomedicines-12-01328-t001:** International definition of iMCD-TAFRO.

1. Definite iMCD-TAFRO Criteria
1.1 Clinical Criteria (all four required)
Thrombocytopenia (T): Pre-treatment nadir platelet level ≤ 10 × 10^4^/µL; Anasarca (A): Pleural effusion, ascites, or subcutaneous edema with CT scan; Fever or hyperinflammatory status (F): Fever ≥ 37.5 °C of unknown etiology or CRP ≥ 2.0 mg/dL; Organomegaly (O): Small volume lymphadenopathy in two or more regions, hepatomegaly, or splenomegaly on CT scan.
1.2 Pathological Criteria (required)
Lymph node consistent with iMCD: Must be consistent with histopathologic features of the international iMCD diagnostic criteria [15] In brief, atrophic germinal centers, concentric rings of mantle zone cells, and interfollicular hypervascularization or plasmacytosis. Negative for light chain restriction and HHV-8.
1.3 Additional Clinical and Pathological Criteria (at least one of the following required)
Renal insufficiency (R): Pre-treatment eGFR ≤ 60 mL/min/1.73 m^2^, creatinine > 1.1 mg/dL (female)/ > 1.3 mg/dL (male), or renal failure necessitating hemodialysis; TAFRO-consistent bone marrow: Reticulin fibrosis (R) or megakaryocytic hyperplasia, without evidence of an alternative diagnosis.
1.4 Exclusion Criteria (required): see below
1.5 Supportive Clinical Criteria (not required but strongly supportive)
Renal insufficiency (R): Pre-treatment eGFR ≤ 60 mL/min/1.73 m^2^, creatinine > 1.1 mg/dL (female)/ >1.3 mg/dL (male), or renal failure necessitating hemodialysis; TAFRO-consistent bone marrow: Reticulin fibrosis (R) or megakaryocytic hyperplasia, without evidence of an alternative diagnosis; Absence of polyclonal hypergammaglobulinemia (immunoglobulin G ≤ 1.2× upper limit of normal by nephelometry; Elevated alkaline phosphatase with mild to no elevation in bilirubin and transaminases.
2. Probable iMCD-TAFRO Criteria: All four Clinical Criteria and Additional Clinical and Pathological Criteria met, but Pathological Criteria not able to be assessed because no lymph node biopsy was performed or an insufficient specimen was obtained.
3. TAFRO syndrome, not iMCD-TAFRO: All four Clinical Criteria and Renal insufficiency (R) met, but lymph node biopsy was not consistent with iMCD or an exclusion criteria diagnosis was made.
Exclusion Criteria—Must exclude the following diseases
Infectious diseases—including the below but not limited to: HHV-8; EBV-associated lymphoproliferative disorders; Acute HIV infection; Tuberculosis; COVID-19 cytokine storm syndrome.
Autoimmune/rheumatologic diseases: Systemic lupus erythematosus; Sjögren syndrome; Rheumatoid arthritis; Adult-onset Still disease; Juvenile idiopathic arthritis; IgG ≥ 3400 mg/dL (suggestive of autoimmune diseases or plasma cell dyscrasias); Primary hemophagocytic lymphohistiocytosis.
Malignancy—including the below but not limited to: Malignant lymphoma; Multiple myeloma; Metastatic cancer; POEMS syndrome.

Abbreviations: CRP, C-reactive protein; CT, computed tomography; eGFR, estimated glomerular filtration rate; HHV-8, human herpesvirus 8. Reproduced with permission from Nishimura, Y., Fajgenbaum D.VC., Pierson S.K. et al. Validated International Definition of the TAFRO clinical subtype of idiopathic multicentric Castleman disease. Am J Hematol, 2021, 96(10), 1241–1252 [21].

**Table 2 biomedicines-12-01328-t002:** Comparison of iMCD-TAFRO and TAFRO syndrome criteria/definition to date.

Inclusion Criteria			Exclusion Criteria
Required Histopathological Criteria	Required Clinical Criteria	Other Criteria	
Not specified (noted in minor criteria)	(Mandatory) Anasarca, including pleural effusion, ascites, and general edema Thrombocytopenia (≤10 × 10^4^/µL) Systemic inflammation^42^	(Need 2 or more) Castleman disease-like features on LN Reticulin myelofibrosis and/or hyperplasia of megakaryocytes in BM Mild organomegaly Progressive renal insufficiency	Malignancies Autoimmune disorders Infectious disorders POEMS syndrome Cirrhosis TTP/HUS
(Mandatory for definite diagnosis) LN that is consistent with the histopathological features of the international iMCD diagnostic criteria [15]	(Mandatory) Thrombocytopenia: Minimum pre-treatment platelet ≤ 10 × 10^4^/µL Anasarca (Pleural effusion, ascites, or subcutaneous edema) Fever or hyperinflammatory status: Fever ≥ 37.5 °C of unknown etiology or CRP ≥ 2.0 mg/dL Organomegaly: Small volume lymphadenopathy (≤2.0 cm) in more than two regions, hepatomegaly, or splenomegaly on CT	(Not required, but contributory)Renal insufficiency: Pre-treatment eGFR ≤ 60 mL/min/1.73 m^2^, creatinine > 1.1 mg/dL (female)/ > 1.3 mg/dL (male), or renal failure necessitating hemodialysis. Absence of polyclonal hypergammaglobulinemia (immunoglobulin G ≤ 1.2× upper limit of normal by nephelometry High ALP with mild to no increase in bilirubin and transaminases Hyperplasia of megakaryocytes or reticulin fibrosis in BM	Infectious diseases HHV-8, EBV, HIV, TB, COVID-19-CSS Autoimmune/rheumatologic diseases SLE, SjS, RA, AOSD, JIA, IgG ≥ 3400 mg/dL, HLH Malignancy ML, MM, metastatic cancer, POEMS syndrome

Adapted from Nishimura et al. [21]. Abbreviations: ALP, alkaline phosphatase; AOSD, adult-onset Still disease; BM, bone marrow; COVID-19-CSS, COVID-19 cytokine storm syndrome; CRP, C-reactive protein; CT, computed tomography; EBV, Epstein–Barr virus; eGFR, estimated glomerular filtration rate; ESR, erythrocyte sedimentation rate; HHV-8, human herpesvirus-8; HIV, human immunodeficiency virus; IgG, immunoglobulin G; JIA, juvenile idiopathic arthritis; LN, lymph node; ML; malignant lymphoma; MM, multiple myeloma; RA, rheumatoid arthritis; SjS, Sjögren syndrome; SLE, systemic lupus erythematosus; TB, tuberculosis; TTP/HUS, thrombotic thrombocytopenic purpura/hemolytic uremic syndrome.

## Data Availability

Data are contained within the article.

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
