# Peer review of "Increase in Vascular Endothelial Growth Factor (VEGF) Expression and the Pathogenesis of iMCD-TAFRO"

_biomedicines, 2024, doi:10.3390/biomedicines12061328_

Round 1

Reviewer 1 Report

Comments and Suggestions for Authors

The authors present a review on the role of VEGF in the pathology of iMCD-TAFRO. Regarding iMCD-TAFRO, a group of diseases for which the concept of disease is not completely defined, I get the impression that the content is very easy to understand from the introduction, but the explanation using diagrams about VEFG is redundant. I think there is a lot of waste, so I think it should be simplified further.

Author Response

Please view the attached file.

Reviewer 2 Report

Comments and Suggestions for Authors

In the manuscript titled "Role of Vascular Endothelial Growth factors (VEGF) in the pathogenesis of iMCD-TAFRO," Srkalovic et al. explore the potential role of VEGF in the development of iMCD-TAFRO and the possibility of targeting related signaling pathways. The topic is intriguing, and the authors have thoroughly reviewed the molecular events underlying iMCD-TAFRO. The manuscript is well-written, and the tables and figures are informative. However, the figure legends of figures 2 and 5 are brief, and the authors should provide a more detailed description of the figures. Overall, the manuscript is scientifically sound.

Author Response

Please view the attached file.

Reviewer 3 Report

Comments and Suggestions for Authors

The review proposed by the authors for publication in Biomedicines Journal is written in substantially correct English and provides the reader with a broad overview of the covered research topic. The work aims to describe a possible role of Vascular Endothelial Growth Factor in pathophysiology of the complex iMCD-TAFRO disorder. The title of this review is in fact “Role of Vascular Endothelial Growth factors (VEGF) in the pathogenesis of iMCD-TAFRO”. Actually, in their manuscript, Srkalovi  et al summarize the results from previous works in which increased levels of VEGF-A expression related to TAFRO syndrome are found, monitored in patients affected by the pathology. Furthermore, they underline the synergistic effect of VEGF/IL-6 axis based on the increased levels experimentally observed for both the cytokines in clinical manifestations of the disease. However, the authors not elucidate a real role for VEGF in TAFRO syndrome, because they themselves state that growth factor “function” is actually ambiguous and controversial in these settings, especially in renal injury associated to TAFRO, and that the molecular mechanisms involving VEGF in phenotypic manifestations found in affected patients are not well known. Therefore, I would like to suggest to the authors to change the title of the review in such a way as not to stress the concept of the “role” of VEGF, but rather to underline the increase of VEGF expression levels correlated to the pathology.

Minor revisions

_Just a suggestion: All the sub-types of the pathology described by the authors in the introduction section results at times in a burdening of the text and distracts the reader due to the amount of informations which are not all necessary. In particular, the introduction paragraph could be lightened by removing the description of the sub-types of iMCD-NOS, given that the authors then focus on iMCD-TAFRO.

_The indication of Table 1 in the text is not clear. On page 5 of the manuscript the title "Table 1. International Definition of iMCD-TAFRO" appears above the table. On page 6 there is a caption under which the table continues. Please the authors move the caption with the specification of the abbreviations to the end of Table 1, otherwise it may seem that they are two different tables. Furthermore, why in the table 1 the authors inserted the line: “1.4 Exclusion Criteria (required): see below”, putting the exclusion criteria on page 6? I strongly suggest to the authors to add exclusion criteria immediately after point 1.4.

_Line 284: Please the authors replace “elevated VEGF”, with “elevated VEGF levels”

_Lane 318: Please the authors specify in the text the ULN acronym (Upper Limit Normal)

Author Response

Please view the attached file.
